# Sowing Date Affects Maize Development and Yield in Irrigated Mediterranean Environments

**Angel Maresma, Astrid Ballesta, Francisca Santiveri and Jaume Lloveras ***

Agrotecnio Center, University of Lleida, Rovira Roure 191, 25198 Lleida, Spain;
angel.maresma@pvcf.udl.cat (A.M.); astrid@hbj.udl.es (A.B.); santiveri@pvcf.udl.cat (F.S.)
* Correspondence: Jaume.lloveras@udl.cat

**Abstract:** Timely sowing is critical for maximizing yield for both grain and biomass in maize. The effects of early (mid-March), normal (mid-April), and late (mid-May) sowing date (SD) were studied over a three-year period in irrigated maize under Mediterranean conditions. Early SD increased the number of days from sowing to plant emergence. Late SD reduced the number of days to plant maturity, and had higher forage yields, higher grain humidity, and taller plants. The average grain and forage yields achieved were 13.2 and 21.3 Mg ha$^{-1}$; 14.0 and 25.1 Mg ha$^{-1}$; and 12.8 and 27.6 Mg ha$^{-1}$, for crops with early, normal, and late SD, respectively. The data support the general perception of farmers that April sowings are the most appropriate in the area where the experiments were carried out. Early SD resulted in lower population densities, while later SD did not yield (grain) as high. However, late SD produced taller plants that contributed to achieve higher forage yields. Late SD could be interesting for double annual forage cropping systems. Sowing at the most appropriate time, when the soil is warm, ensures a good level of maize grain production. Future research could focus in the effect of SD for total annual yields in double-annual cropping systems.

**Keywords:** corn; forage yield; grain yield; plant height; planting; population density; sowing date

---

**Highlights:** Sowing date affects the average maize grain and forage yields. Germination and population density was reduced in mid-March sowing date. Traditional sowing date (mid-April) achieved highest grain yields. Mid-May sowing date was the most appropriate for forage production.

## 1. Introduction

Timely sowing is critical for maximizing yield for both grain and biomass in maize [1,2] and therefore, growers are concerned about the yield response of maize to sowing date (SD) [3–5]. However, optimum maize SD may vary from area to area due to differences in climate and the length of the growing season where the crop is produced [6].

It is known that maize needs warm soil to germinate and grow [5,7]. However, the practice of sowing as soon as possible to take advantage of the solar radiation [8] is nowadays more adopted by farmers. Early planting could contribute to the profitability of maize by increasing yields (crop has more time to photosynthesize) and, in some areas, by avoiding artificial grain drying at the end of the cycle [9–11].

Breeding programs have facilitated germination of maize at colder temperatures [12,13]. Bruns and Abbas [6] reported technological improvements in maize hybrids such as better early season vigor and tolerance to germination in cool wet soils, better seed treatments to guard against damping off diseases and seedling insect pests, or the advent of herbicides. These factors have contributed to planting maize earlier than it was 30 years ago [5].

In general, early sowing is preferable, but temperatures must be high enough to ensure quick germination and emergence. Also, SD must be late enough to avoid late spring frosts. As a rule, maize

should not be sown until the soil temperature approaches 10 °C. Under cold soil conditions (below 10 °C), seeds will readily absorb water but will not initiate root or shoot growth, which leads to seed rot and poor emergence [5,14].

Increases in temperature during the vegetative period of maize crops hastens the growth rate more than the development rate, resulting in taller plants with a larger biomass [15]. Thus, under field conditions, rising temperature reduces the duration of crop growth, and consequently SD reduces the time during which incident radiation can be intercepted and transformed into dry matter (DM) [16].

The highest yields generally occur where the growing season is longest and soil moisture is not a limiting factor [17]. Yield reductions due to early or late planting have been well documented in the literature [4,5,9,11,16,18]. Early planting results in reduced cumulative intercepted photosynthetically active radiation (IPAR) because of delayed leaf area development. High temperatures under late planting scenarios also reduce cumulative IPAR by reducing the calendar time for crop development, and thereby, decreasing yields [19].

Optimum SD vary from one environment to another [8]. In the Ebro valley, the month of April is the most recommended sowing period for maize, particularly, the first half of the month [20–22]. Even so, there can be year-to-year variations associated with temperatures and rainfall during spring. In Mediterranean areas, maize is often grown under irrigation. Thus, crop growth depends on water availability during the growing season, and in some areas on irrigation turns. Many farmers currently practice double-annual cropping of maize after winter forage in order to increase the economic viability of their farms [23]. Therefore, in Mediterranean environments, a large variability exists in maize planting date, and there is a need to quantify the effect of the SD on maize yield.

The objective of this research was to evaluate the effects of the date of sowing (early, normal, and late) on maize yields (grain and forage) and crop growing period, in irrigated Mediterranean environments.

## 2. Materials and Methods

A three-year experiment (2003–2005) was conducted at the IRTA experimental station at Gimenells, Catalonia, Spain (41°65′ N, 0°39′ E), under sprinkler irrigated conditions. The study area is characterized by a semi-arid climate with low annual precipitation (345 mm) and a high annual average temperature (14.6 °C) (Table 1).

**Table 1.** Mean monthly ($T_m$) air temperatures and total monthly rainfall at Gimenells, during the experiment (from 2003 to 2005). Long-term (30 year) mean annual temperature and rainfall values at Gimenells are 14.6 °C and 345 mm, respectively.

| Month | 2003 | | 2004 | | 2005 | |
|---|---|---|---|---|---|---|
| | $T_m$ | Rainfall | $T_m$ | Rainfall | $T_m$ | Rainfall |
| | (°C) | (mm) | (°C) | (mm) | (°C) | (mm) |
| February | 5.7 | 70.6 | 4.7 | 50.6 | 4.1 | 8.6 |
| March | 10.9 | 30.5 | 8.3 | 37.4 | 9.3 | 9.2 |
| April | 13.4 | 25.9 | 11.6 | 61.1 | 13.7 | 7.2 |
| May | 17.6 | 62.5 | 15.8 | 40.8 | 18.4 | 53.1 |
| June | 24.9 | 15.1 | 22.6 | 7.6 | 23.2 | 13.1 |
| July | 25.1 | 2.0 | 23.1 | 49.8 | 24.5 | 18.5 |
| August | 25.9 | 38.0 | 23.7 | 6.0 | 22.4 | 20.9 |
| September | 19.3 | 101.5 | 20.6 | 18.5 | 19.3 | 28.6 |
| October | 13.9 | 71.2 | 16.1 | 31.0 | 15.8 | 82.0 |

The soil was a Petrocalcic Calcixerept [24], which is representative of many areas of the Ebro valley (Table 2). Two maize cultivars with different growth cycles were sown: Cecilia (600 FAO) and Eleonora (700 FAO). The cultivars provided good representations of the 600 to 700 FAO cycles, which are the ones most commonly used in the area [20].

**Table 2.** Soil properties at the beginning of the experiment (2003).

| Soil Properties | Horizon | | |
|---|---|---|---|
| | Ap 0–25 cm | Bwk$_1$ 25–70 cm | Bwk$_2$ 70–120 cm |
| Sand, % | 38.5 | 38.4 | 44.6 |
| Silt, % | 40.3 | 41.9 | 38.4 |
| Clay, % | 21.2 | 19.7 | 17.0 |
| pH | 8.1 | 8.2 | 8.3 |
| Organic matter, g kg$^{-1}$ | 22.0 | 14.0 | 6.2 |
| EC$_{1:5}$, dS m$^{-1}$ | 0.20 | 0.34 | 0.59 |
| N (N-NO$_3{}^-$), mg kg$^{-1}$ | 33 | - | - |
| P (Olsen), mg kg$^{-1}$ | 38 | 20 | 10 |
| K (NH$_4$Ac), mg kg$^{-1}$ | 241 | 94 | 59 |

The statistical design of the maize experiments was a split plot, with four replications, where SD were used as the main plot, and cultivars as the subplots [25]. For each year, the different plots (harvest date) and subplots (cultivars) were randomly distributed.

Maize was sown at three different dates, which were as close as possible to 15 March, 15 April, and 15 May. The exact dates are presented in Table 3.

**Table 3.** Dates of sowing and harvesting for maize. The sowing rate was 85,000 plants ha$^{-1}$ with a distance of 71 cm between rows. The plot size of each experimental plot was of 15 × 11 m.

| Year | Sowing Date | | | Harvest Date | |
|---|---|---|---|---|---|
| | 1 | 2 | 3 | Biomass | Grain |
| 2003 | 27 March | 14 April | 14 May | 16–29 September | 1 October |
| 2004 | 15 March | 14 April | 17 May | 8–22 September | 5 October |
| 2005 | 14 March | 14 April | 19 May | 12 September | 10 October |

Conventional tillage was carried out before sowing. This included disc ploughing and cultivation to a depth of 30 cm to incorporate previous stover and to prepare the soil for the sowing. The maize was fertilized with 50 kg N ha$^{-1}$, 150 kg ha$^{-1}$ of P$_2$O$_5$ and 200 kg ha$^{-1}$ K$_2$O before sowing, and then two equal side dressings of 100 kg N ha$^{-1}$ were applied at V4–V5 and at V6–V7 maize growing stages. Nitrogen was applied as ammonium nitrate (34.5% N), and maize was irrigated after the application to avoid N losses. In each growing season, around 650 mm of irrigation water were applied to the maize crop. Irrigation water was of good quality and did not contain any significant amounts of nitrates.

A pre-emergence herbicide (1 L ha$^{-1}$ 96% metolachlor and 3 L ha$^{-1}$ 47.5% atrazine) was applied to control weeds. When necessary hand wedding or post-emegence herbicide were applied to control *Abutilon theophrasti* (Banvel (20% fluoxypyr) at a rate of 1 L ha$^{-1}$), and to control Sorghum *halepense* (Elite (nicosulfuron 4%) at a rate of 1.5 kg ha$^{-1}$).

The height of 10 plants of the central rows was measured about one week after silking in each plot from the base of the crop to the last leaf. At same time, the leaf are index (LAI) was measured by taking all the leaves of five consecutive plants from one central row and measuring them with the LAI meter (Li-Cor, Lincoln, NE, USA). Intercepted solar radiation was measured by taking eight readings per plot from the central rows, at noon, using a Ceptometer (Delta-T devices, Burnell, UK). The plant density was estimated before forage harvest, counting the total plants of the two central rows in 5 m strips (1.42 m by 5 m).

The forage and grain harvest took place during September or October, after the plants had reached physiological maturity (Table 3). Grain yield was measured by harvesting two central rows (1.42 m by 15 m) from each plot using an experimental plot combine. Grain moisture was determined from a 300-g grain sample taken from each plot, using a GAC II (Dickey-John, Auburn, IL, USA), and the grain

yield was adjusted to 14% moisture. The aboveground biomass yield was determined at physiological maturity by harvesting plants from one central row (0.71 m by 5 m) at ground level. Subsamples were chopped and dried in a stove at 65 °C for at least 48 h to determine the DM weight.

A mixed-effects analysis of variance (ANOVA) was carried out to assess the responses to SD, with years evaluated as repeated measurements.

## 3. Results and Discussion

### 3.1. Duration of the Growing Cycle

The duration of the growing cycle (number of days from sowing to physiological maturity) decreased with each delay in sowing, falling from an average of 162 days with the earliest sowings (mid-March), to 143 and 125 days with the sowings on mid-April and on mid-May (Table 4), respectively.

The average number of days from sowing to plant emergence in the mid-March SD was 22, and was reduced to 12 and 9 days for the mid-April and mid-May SD, respectively. However, the greatest effect of delaying sowing was observed in the number of days from sowing to silking, which fell significantly, from 104 days in the mid-March SD, to 81 days (22% reduction) and to 69 days (33% reduction) for the mid-April and mid-May SD (Table 4). This is in agreement with Mederski and Jones [26], who reported a decrease in the number of days from sowing to silking as soil temperature increases. Moreover, the number of days from emergence to silking was considerably reduced in our study. It ranged from 82 to 60 days from the first to the last SD. There was a cultivar effect in the length of the growing cycle and the time to silking (Table 4). However, the time from sowing to emergence was similar in both cultivars. Eleonora (700 FAO) required 87 and 147 days from sowing to silk and to physiological maturity, respectively. Cecilia (600 FAO) required on average 4 and 6 days less than Eleonora, which represent about a 4% of the total time to arrive to each growing stage. These findings could be expected due to the different growing cycle of the cultivars.

Soils and air temperature were the main reason for these differences in growth duration. Warmer temperatures accelerate the rate of crop development, resulting in shorter vegetative and reproductive phases [26,27]. Although these differed from year to year because of annual variations in temperature (Table 1), growth duration clearly decreased when sowing was delayed. The average temperature in March only reached 10.9 °C in the first year of the experiment, whereas in the last two years it barely reached 9 °C. In the second year the average temperature in March was 8.3 °C, which is considered low for maize sowings and which occasioned a large period from sowing to emergence (28 days). Average April temperatures were not very high either. They were above 10 °C, but never exceeded 13.7 °C. During June, July and August, the average monthly temperatures ranged from 22.4 °C to 25.9 °C (Table 1).

**Table 4.** Sowing date (SD) (mid-March, mid-April, and mid-May) effect on average forage maize dry matter (DM) yield at maturity, grain yield and humidity, the dates of emergence, silk and black layer appearance, and the days happened between sowing and silk and black layer appearance, plant height, leaf area index (LAI) and intercepted solar radiation (measured one week after silking). Average 2003–2005.

| Sowing Date (SD) | Forage Yield | Grain Yield (14% hum.) | Harvest Index | Grain Humidity | Emergence | Silk | Black Layer | Plant Height | Plant Density | LAI | Intercepted Solar Radiation |
|---|---|---|---|---|---|---|---|---|---|---|---|
| | (Mg ha$^{-1}$) | (Mg ha$^{-1}$) | | (%) | (days) | (days) | (days) | (m) | (plants ha$^{-1}$) | (m$^2$ m$^{-2}$) | (%) |
| **Mid-March** | **21.3** | **13.2** | **0.60** | **15.7** | **22** | **104** | **162** | **2.21** | **70,764** | **3.57** | **84.9** |
| 2003 | 22.0 | 12.5 | 0.57 | 16.5 | 20 | 93 | 144 | 2.25 | 73,541 | 4.30 | 82.8 |
| 2004 | 22.4 | 14.2 | 0.63 | 16.9 | 28 | 111 | 178 | 2.15 | 80,000 | 3.02 | 89.1 |
| 2005 | 19.6 | 13.0 | 0.66 | 13.9 | 18 | 108 | 164 | 2.24 | 78,333 | 3.39 | 82.7 |
| **Mid-April** | **25.1** | **14.0** | **0.55** | **16.8** | **12** | **81** | **143** | **2.33** | **78,333** | **4.02** | **88.7** |
| 2003 | 25.4 | 13.3 | 0.52 | 17.8 | 10 | 78 | 140 | 2.34 | 70,000 | 4.20 | 87.9 |
| 2004 | 27.8 | 15.7 | 0.56 | 17.7 | 14 | 84 | 152 | 2.41 | 80,833 | 4.12 | 95.1 |
| 2005 | 22.0 | 12.9 | 0.59 | 15.0 | 12 | 82 | 137 | 2.25 | 75,833 | 3.74 | 83.2 |
| **Mid-May** | **27.6** | **12.8** | **0.45** | **24.9** | **9** | **69** | **125** | **2.53** | **75,069** | **4.88** | **91.9** |
| 2003 | 27.6 | 11.5 | 0.42 | 27.5 | 9 | 68 | 126 | 2.61 | 68,750 | 5.50 | 96.7 |
| 2004 | 30.7 | 15.4 | 0.50 | 25.0 | 10 | 72 | 129 | 2.59 | 74,166 | 4.88 | 94.9 |
| 2005 | 24.5 | 11.5 | 0.47 | 22.4 | 8 | 66 | 121 | 2.38 | 71,042 | 4.26 | 84.0 |
| ANOVA | | | | | | | | | | | |
| Year (Y) Error | * | ** | * | ** | ** | ** | ** | ** | ** | ** | ** |
| Sowing Date (SD) | ** | ** | ** | ** | | ** | ** | * | ** | ** | ** |
| SD*Y Error | * | * | * | ** | ** | ** | ** | ** | * | ns | * |
| Cultivar (C) | ns | ns | ns | ** | ns | ** | ** | * | ns | ** | ** |
| C*SD | ns | * | ns | ** | ns | * | * | ns | ns | ns | ns |
| Y*C | ns | ns | ns | ns | ns | ns | ** | ns | ns | * | * |
| Y*C*SD | ns | ns | ns | ns | ns | ** | ** | ns | ns | ns | ns |

* Significant at *p*-value < 0.05; ** Significant at *p*-value < 0.01; ns = not significant.

### 3.2. Plant Height, LAI, Intercepted Solar Radiation, and Plant Density

As previously described by other authors [1,5,28], plant height increases with delayed sowing. In the present experiment, maize plants height increased significantly (13%) from an average of 2.21 m for the mid-March SD to 2.53 m for plants sown on May (Table 4). Warm weather during early vegetative growth can stimulate plants to develop larger vegetative structures [16]. Despite cultivars showed differences in plant height, in the three-year average plant height was 2.40 m for Eleonora and 2.32 m for Cecilia. Those differences in plant height were smaller than the observed between the SD. There were significant differences in the LAI. The taller plants of Eleonora contributed to obtain higher LAI and intercepted solar radiation than Cecilia. The LAI and the intercepted solar radiation were respectively 0.4 $m^2$ $m^{-2}$ and 3% higher for Eleonora. Late SD averaged LAI values of 4.88 $m^2$ $m^{-2}$, whereas the SD of mid-March and mid-April averaged respectively 3.57 and 4.02 $m^2$ $m^{-2}$. Thus, the tallest plants were also the ones that obtained the highest LAI values. Consequently, the amount of intercepted solar radiation was also higher for the late SD (Table 4). The LAI indexes obtained in our trials were lower than those reported by Tsimba et al. [4] in New Zealand. However, the grain yields achieved were similar.

The amount of intercepted solar radiation differed depending on the growth periods associated with the different sowings. Probably, this fact could help to explain the differences in yield and DM content in the three SD (Table 4). As reported by Cirilo and Andrade [16], late sowings resulted in high crop growth rates during the vegetative period because of high radiation use efficiency (RUE) and high percent radiation interception. However, these treatments resulted in low crop growth rates during grain filling because of low RUE and low incident radiation.

March temperatures in the second and third growing seasons were below 10 °C (Table 1), which affected the germination of the maize in the early SD. In the second growing season, there were 28 days between sowing and emergence for the mid-March SD. This was a long period and some of the plants did not emerge at all. Only 70,000 of the 85,000 plants $ha^{-1}$ initially sown emerged (with plant losses of about 17%). Therefore, poor maize germination is one of the possible consequences of early sowing in some years. Moreover, the slow growth of maize made it less competitive with weeds and more weed control was necessary. Other researchers [14] have also reported these kinds of results. Earlier sowing may not be the most interesting option, whereas sowing at the appropriate time when the soil is warm, tends to ensure a good plant stand. Both, year and SD were significant for the total plant density. Temperature plays an important role for the successful development of seed to plants, but others factors such as soil preparation or precipitation can influence the germination. The lowest plant density was determined for 2005, which had the driest early season for the studied fields. The final three-year average densities for the three SD were around 71,000 plants $ha^{-1}$, 78,000 plants $ha^{-1}$ and 75,000 plants $ha^{-1}$, for the early, middle, and late SD, respectively.

The differences in LAI observed seem to confirm the need to increase the sowing density at early SD in order to achieve sufficient photosynthetically active radiation interception [4]. As previously mentioned, the number of days from sowing to silking decreases with increases in soil and air temperature. Indeed, temperature has a major influence on the rate of maize development [4,7,16,29]. According to Duncan [7], the rate of maize development, from sowing to anthesis, is a function of temperature more than of photosynthesis.

Photoperiod can also influence maize development and grain maturation [30,31]. However, the differences in photoperiod associated with the sowing periods considered in this study were not sufficiently large at the critical photoperiod-sensitive interval (at tassel initiation or at between stages V5 and V7) to have had an impact on plant development [30]. Past studies indicate that differences in photoperiod from 3 h to 5 h are needed during the photoperiod-sensitive interval to generate differences in the phenological response of the Corn Belt germplasm [31].

On the first year of the experiment, there were storms and strong winds few days before harvesting, and as a result, many plants were lodged. Around the 21–23% of the plants from the mid-March, and mid-April sowings were lodged, whereas the 74% of the plants from the last sowing (mid-May) were

affected. The difference among SD could probably be affected by the height of the plants. The more optimal growing conditions (mainly temperature) for the last SD contributed to increase the height of the plant and made them more vulnerable to lodging when storms occur at the end of the crop cycle. In the other years, lodging was inconsequential.

### 3.3. Grain and Biomass Yields

The optimum SD for grain and biomass were similar in all of the studied years despite of some year-to-year variation (data not shown). Maize sown in mid-April achieved the highest average grain yields (14.0 Mg ha$^{-1}$), followed by mid-March sowings (13.2 Mg ha$^{-1}$) and the lowest grain yields were achieved with mid-May sowings (12.8 Mg ha$^{-1}$). Every year, mid-April SD yielded higher than the mid-March and mid-May alternatives, except for the last year where the mid-March SD achieved similar yields. That yield variability among SD (Table 4) was expected because of the different weather conditions of the experiment each year (year and SD*year were significant). Mid-April is the most common SD in the studied area, possibly because of the cooler temperatures during mid-March SD and the shorter growing season associated with mid-May SD. The average grain yield obtained in the study is similar to the averages reported by Cela et al. [32] (13.6 ± 0.4 Mg ha$^{-1}$) in the same area.

Grain humidity at harvest time increased with delays in sowing, varying from 24.9% at the last SD (mid-May) to 16.8% and 15.7% to the mid-April and mid-March SD, respectively. Cultivar also had an effect in grain humidity. The longer growing cycle of Eleonora (700 FAO) was translated into higher grain humidity at harvest (20.8%) compared with Cecilia (17.5%). This may prove important in a few years, because the drying of the grain increases the production costs and consequently the maize profitability.

Forage yield increased when delaying the SD, similarly to the results reported by Bunting [1], Dillon and Gwin [28], and Fairey [33]. Indeed, Mederski and Jones [26] reported that increasing soil temperature accelerates the rate of development of maize and produce significant increases in DM production. Although, in some conditions there were reported no differences when delaying the SD [34].

The highest biomass yields (27.6 Mg ha$^{-1}$) were associated with the latest sowings (May): in which the plants grew taller, although with less grain proportion than the earlier SD. The harvest index decreased from 0.6, in early sowings, to 0.54 in mid-period sowings and to 0.44 for late sowings (Table 4). This fact may be interesting for forage production farmers who use double cropping systems. They can grow a forage crop during winter and thereafter plant maize in summer, which have higher forage yield potential than growing a monocropped maize [23]. However, a quality analysis of the forages may be required as Deinum and Struik [35] and Bunting [1] suggested that delaying sowing may reduce forage digestibility because of the lower grain proportion.

Cirilo and Andrade [16] reported that crop DM partitioning was strongly affected by SD. Early sowing favored reproductive growth, whereas late sowing favored vegetative growth. Delays in the SD hastened plant development between seedling emergence and silking, reducing crop exposure to cumulative incident radiation during the vegetative period. Dobben [15] indicated that increases in temperature during the maize vegetative period hastened growth rate more than development rate, resulting in taller plants with larger biomasses.

Sowing date has a significant effect on maize grain yield when all other factors are equal. Research across the USA has shown that there is an 'ideal' sowing window, with a decline in grain yield with each additional day after it, as less light and fewer growing degree days are available to the plant [5,6,11]. However, this 'ideal' sowing window is not constant over the years and may vary according to the weather, as it happened in one out of three years of the study (third year), in which although without significant differences, the first SD obtained slightly higher grain yields (Table 4).

The influence of SD and plant density on maize grain yield may be related to IPAR and LAI. In Argentina, delayed SD have been shown to increase IPAR levels at the silking stage by increasing

leaf area development as a result of higher temperatures during vegetative growth [16]. Even so, yields were still lower with delayed SD due to reduced levels of cumulative IPAR.

## 4. Conclusions

The duration of the growing cycle (number of days from sowing to physiological maturity) was reduced by each delay in sowing. Early sowings increased the period from sowing to plant emergence, which reduced the germination and the population density of the crop. Alternatively, late sowings reduced the number of days to physiological maturity producing higher humidity content in grain at harvest and taller plants.

Sowing dates of middle April seem to be the most interesting for achieving the maximum grain yields under irrigated Mediterranean conditions. To anticipate the SD may not always be interesting; it will depend on soil temperature of the year to ensure germination. However, if the interest is on the forage yields (albeit with a lower proportion of grain yield), sowing can be delayed in order to benefit from the higher growth of the maize at higher temperatures, as well as to open the window to grow a winter crop.

**Author Contributions:** Á.M. contributed in the analysis of the data and in the writing of the bulk of the paper. A.B. and F.S. acquired field data. J.L. conceived and designed the experiment, and contributed in the analysis of the data and in the writing of the bulk of the paper.

**Funding:** This work was supported by the DARP (Department of Agriculture, Livestock, Fishing and Food) of the Generalitat de Catalunya.

**Acknowledgments:** The authors would like to thank the students and technicians, Josep Pons, Pau Marcé, Albert Casals, Silvia Martí form UdL and Josep Anton Betbesé and Jose Luis Millera of the IRTA for their help with the sowing and harvesting of the crops.

**Conflicts of Interest:** The authors declare no conflict of interest. The funding sponsors had no role in the design of the study; in the collection, analyses, or interpretation of data; in the writing of the manuscript, or in the decision to publish the results.

## Abbreviations

| | |
|---|---|
| ADF | acid detergent fiber |
| CP | crude protein |
| DM | dry matter |
| LAI | leaf area index |
| NDF | neutral detergent fiber |
| SD | sowing date |

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
