# Peer review of "Sowing Date Affects Maize Development and Yield in Irrigated Mediterranean Environments"

_agriculture, doi:10.3390/agriculture9030067_

Round 1
Reviewer 1 Report
The manuscript focus in the study of the effect of different sowing dates on maize development and yields, grain and forage, for two different FAO cycles cultivars. The 3 years duration of the experiment, gives soundness to the results obtained.
The article give relevant information as optimal sowing date depend not only of cultivar but also on each specific environment.
Only two concerns. First, the study was developed in years 2003-2005, and genetic improvement move fast. How representative are the selected hybrids in comparison to the new maize hybrids in 2018?. .
Second, in Table 4 some interactions between hybrids and SD are significant but only information of SD or SD and years is given in the Results and Discussion section. I think if would be of interest to include some information on the two hybrids if they behave different.
ABSTRACT
Line 15
Early SD resulted is lower population..
Early SD resulted in lower population
Lines 15 -16
..while later SD did not yield as high (grain)
.. while later SD did not yield (grain) as high
Line 17
for forage double annual cropping systems..
for double annual forage cropping systems ..
INTRODUCTION
The introduction section is well organized, and it provides relevant information to focus the interest in the objectives of the manuscript.
Only one concern in lines 55 -57
Early planting results in reduced intercepted photosynthetically active radiation (IPAR) because of delayed leaf area development. However, high temperatures under late planting scenarios also reduce IPAR by reducing the calendar time for crop development, and thereby, decreasing yields.
I do not understand these two phrases, IPAR is a variable that is measured at specific times, and thus the timing of IPAR measurement should be given. IPAR is lower at the beginning of crop development and then increase to a maximum for fully vegetative development. Perhaps the authors means cumulative IPAR?. Please specify.
Line 47
period of maize crops hastened the growth rate
period of maize crops hastens the growth rate
The rest of the paragraph and the section is written in present tense
MATERIALS AND METHODS
- Line 72
2.1 Study area
Please delete the subheading is not relevant.
- Lines 71-121
The material and method section is, from my point of view, a little bit disorganized
I suggest to move up the description of the design of the experiment, to line 81, and reorganize the information,
first experimental design,
second definition of treatments: SD and cultivars and
then agronomical practices
A three-year experiment (2003-2005) was conducted at the IRTA experimental station at Gimenells, Catalonia, Spain (41º65’N, 0º39’E), under sprinkler irrigated conditions. The study area is characterized by a semi-arid climate with low annual precipitation (345 mm) and a high annual average temperature (14.6oC) (Table 1). The soil was a Petrocalcic Calcixerept [24], which is representative of many areas of the Ebro valley. The main soil characteristics are presented in Table 2.
Table 1. Mean monthly (Tm) air temperatures and total monthly rainfall at Gimenells, during the
Experiment
Table 2. Soil properties at the beginning of the experiment. (2003).
The statistical design was a split plot, with four replications, where sowing dates (3) were the main plots, and cultivars (2) the subplots. For each year, the different plots (sowing dates) and subplots (cultivars) were randomly distributed.
Maize was sown at three different dates, which were as close as possible to March 15th, April15th, and May 15th The exact dates are presented in Table 3. Two maize cultivars with different growth cycles were sown: Cecilia (600 FAO) and Eleonora (700 FAO) (Serra et al., 2007). The cultivars provided good representations of the 600 to 700 FAO cycles, which are the ones most commonly used in the area.
Table 3. Dates of sowing and harvesting for maize.
Conventional tillage was carried out before sowing. This included disc ploughing and cultivation to a depth of 30 cm to incorporate previous stover and to prepare the soil for the sowing.
The maize was fertilized with 50 kg N ha-1, 150 kg ha-1 of P2O5 and 200 kg ha-1 K2O before sowing, and then two equal side dressings of 100 kg N ha-1 were applied at V4-V5 and at V6-V7 maize growing stages. Nitrogen was applied as ammonium nitrate (34.5% N), and maize was irrigated after the application to avoid N losses. In each growing season, around 650 mm of irrigation water were applied to the maize crop. Irrigation water was of good quality and did not contain any significant amounts of nitrates.
A pre-emergence herbicide (1 L ha-1 96% metolachlor and 3 L ha-1 47.5% atrazine) was applied to control weeds. When necessary hand wedding and/or and post-emegence herbicide were applied to control Abutilon theophrasti (Banvel, 20% fluoxypyr, at a rate of 1L ha-1), and to control Sorghum halepense (Elite, nicosulfuron 4%, at a rate of 1.5 kg ha-1).
The height of 10 plants ……….
The forage and grain harvest ……………
A mixed-effects analysis of variance (ANOVA) was carried out to assess the responses to SD and cultivars with years evaluated as repeated measurements.
- Line 101
When necessary hand wedding and/or and post-emegence herbicide…
Please change to:
When necessary hand wedding or a post-emegence herbicide …
- Lines 105-107
Specify when LAI and IPAR were measured; at the same time than plant height?
- Lines 115-116
… in a stove at 65 oC for at least 48 hours to determine the DM content
change to
… in a stove at 65 oC for at least 48 hours to determine the DM weight
RESULTS AND DISCUSSION
Lines 128-29
The average number of days from sowing to plant emergence in the mid-March SD was 22, and
was reduced in 10 and 13 days for the mid-April and mid-May SD
better change to
The average number of days from sowing to plant emergence in the mid-March SD was 22, and
was reduced to 12 and 9 days for the mid-April and mid-May SD
I think is more clear to give the information in terms of the days from sowing to emergence for the three SD
Line 143
… growth clearly increased when sowing was delayed.
What do you mean by growth, plant height, forage yield? Please specify better
Lines 152-153
It would be better to give the plant height in meters (international unit), instead in cm. In Table 1 plant heigth is given in meters
Line 158
Consequently, the amount of intercepted solar radiation during their respective periods of growth was also higher for the late SD
IPAR has been measured only once, I suppose that after silking, but no during the period of maize growth. Please, support this phrase, information is not in Table 4. It is form literature?
Line 169
March temperatures in the second and third growing seasons were below 10oC (Table 1), which affected the germination of the maize in the early SD, please add
Lines 177-178
The final three-year average densities for the three SD were 72,000 plants ha-1, 79,000 plants ha-1 and 74,000 plants ha-1, for the early, middle and late SD, respectively.
Please specify in Mat&met how you measure this variable and if possible include the information for the 3 SD and the 2 hybrids in Table 4.
Lines 179-180
The differences in LAI observed seem to confirm the need to increase the sowing density of early maturing hybrids
Data will recommend to increase the sowing density at early SD; there is no information of cultivars.
Please add in that phrase at early SD and remove early maturing hybrids
The differences in LAI observed seem to confirm the need to increase the sowing density at early SD
The hybrids used are early mature hybrids?
Lines 189
differences in photoperiod of from 3 to 5 hours are needed during
Lines 124-149 and Lines 198 on
- Subheading 3.1
In Table 4. There is a significant interaction between V*SD and Y*V*SD in “days to emergence”, “days to silking” and “days to black layer”. You should comment on this if these interactions are qualitative you should consider the analysis for each hybrid separately.
- Subheading 3.3
In addition, there is a significant V*SD interaction for grain yield and grain humidity, you should comment on this

Author Response
Response to reviewer 1
The manuscript focus in the study of the effect of different sowing dates on maize development and yields, grain and forage, for two different FAO cycles cultivars. The 3 years duration of the experiment, gives soundness to the results obtained. The article give relevant information as optimal sowing date depend not only of cultivar but also on each specific environment.
Only two concerns. First, the study was developed in years 2003-2005, and genetic improvement move fast. How representative are the selected hybrids in comparison to the new maize hybrids in 2018?.
Response: The experiment conducted in 2003-2005, but the cultivars used in the experiment (Cecilia (600 FAO) and Eleonora (700 FAO)) have been present in the maize cultivar studies of the area recently. Both cultivars were used as control to evaluate new hybrids (Gutierrez, M., 2014, GENVCE, 2014). Moreover, when we evaluated the yields they were similar to the reported by other authors is new studies. Thus, we thought that the study was relevant and can provide information about sowing dates (which has not been addressed in the study area).
Gutierrez, M. 2014. Results of maize and sunflower variety trials in Aragon (Season 2013). Resultados de la red de ensayos de variedades de maíz y girasol en Aragón. Campaña 2013. Technical report. Government of Aragon.
Second, in Table 4 some interactions between hybrids and SD are significant but only information of SD or SD and years is given in the Results and Discussion section. I think if would be of interest to include some information on the two hybrids if they behave different.
Response: Thanks for your comment. We did not find many differences between the studied hybrids but definitely, we should include in the results and discussion when they differ. We have included that information in the text.
ABSTRACT
Line 15
Early SD resulted is lower population..
Early SD resulted in lower population
Lines 15 -16
..while later SD did not yield as high (grain)
.. while later SD did not yield (grain) as high
Line 17
for forage double annual cropping systems..
for double annual forage cropping systems ..
Response: Thanks for your suggestions. We have modified the text in the manuscript.
INTRODUCTION
The introduction section is well organized, and it provides relevant information to focus the interest in the objectives of the manuscript.
Only one concern in lines 55 -57
Early planting results in reduced intercepted photosynthetically active radiation (IPAR) because of delayed leaf area development. However, high temperatures under late planting scenarios also reduce IPAR by reducing the calendar time for crop development, and thereby, decreasing yields.
I do not understand these two phrases, IPAR is a variable that is measured at specific times, and thus the timing of IPAR measurement should be given. IPAR is lower at the beginning of crop development and then increase to a maximum for fully vegetative development. Perhaps the authors means cumulative IPAR?. Please specify.
Response: You are right, we meant the cumulative IPAR. The study of Otegui et al., (1995) used IPAR measurements at 4 different times. Thanks.
Line 47
period of maize crops hastened the growth rate
period of maize crops hastens the growth rate
The rest of the paragraph and the section is written in present tense
Response: We have modified it. Thanks.
MATERIALS AND METHODS
- Line 72
2.1 Study area
Please delete the subheading is not relevant.
Response: It has been removed. Thanks.
- Lines 71-121
The material and method section is, from my point of view, a little bit disorganized
I suggest to move up the description of the design of the experiment, to line 81, and reorganize the information,
first experimental design,
second definition of treatments: SD and cultivars and
then agronomical practices
A three-year experiment (2003-2005) was conducted at the IRTA experimental station at Gimenells, Catalonia, Spain (41º65’N, 0º39’E), under sprinkler irrigated conditions. The study area is characterized by a semi-arid climate with low annual precipitation (345 mm) and a high annual average temperature (14.6oC) (Table 1). The soil was a Petrocalcic Calcixerept [24], which is representative of many areas of the Ebro valley. The main soil characteristics are presented in Table 2.
Table 1. Mean monthly (Tm) air temperatures and total monthly rainfall at Gimenells, during the
Experiment
Table 2. Soil properties at the beginning of the experiment. (2003).
The statistical design was a split plot, with four replications, where sowing dates (3) were the main plots, and cultivars (2) the subplots. For each year, the different plots (sowing dates) and subplots (cultivars) were randomly distributed.
Maize was sown at three different dates, which were as close as possible to March 15th, April15th, and May 15th The exact dates are presented in Table 3. Two maize cultivars with different growth cycles were sown: Cecilia (600 FAO) and Eleonora (700 FAO) (Serra et al., 2007). The cultivars provided good representations of the 600 to 700 FAO cycles, which are the ones most commonly used in the area.
Table 3. Dates of sowing and harvesting for maize.
Conventional tillage was carried out before sowing. This included disc ploughing and cultivation to a depth of 30 cm to incorporate previous stover and to prepare the soil for the sowing.
The maize was fertilized with 50 kg N ha-1, 150 kg ha-1 of P2O5 and 200 kg ha-1 K2O before sowing, and then two equal side dressings of 100 kg N ha-1 were applied at V4-V5 and at V6-V7 maize growing stages. Nitrogen was applied as ammonium nitrate (34.5% N), and maize was irrigated after the application to avoid N losses. In each growing season, around 650 mm of irrigation water were applied to the maize crop. Irrigation water was of good quality and did not contain any significant amounts of nitrates.
A pre-emergence herbicide (1 L ha-1 96% metolachlor and 3 L ha-1 47.5% atrazine) was applied to control weeds. When necessary hand wedding and/or and post-emegence herbicide were applied to control Abutilon theophrasti (Banvel, 20% fluoxypyr, at a rate of 1L ha-1), and to control Sorghum halepense (Elite, nicosulfuron 4%, at a rate of 1.5 kg ha-1).
The height of 10 plants ……….
The forage and grain harvest ……………
A mixed-effects analysis of variance (ANOVA) was carried out to assess the responses to SD and cultivars with years evaluated as repeated measurements.
Response: The M&M section has been restructured as suggested. Thanks.
- Line 101
When necessary hand wedding and/or and post-emegence herbicide…
Please change to:
When necessary hand wedding or a post-emegence herbicide …
- Lines 105-107
Specify when LAI and IPAR were measured; at the same time than plant height?
- Lines 115-116
… in a stove at 65 oC for at least 48 hours to determine the DM content
change to
… in a stove at 65 oC for at least 48 hours to determine the DM weight
Response: We have included your suggestions.
RESULTS AND DISCUSSION
Lines 128-29
The average number of days from sowing to plant emergence in the mid-March SD was 22, and
was reduced in 10 and 13 days for the mid-April and mid-May SD
better change to
The average number of days from sowing to plant emergence in the mid-March SD was 22, and
was reduced to 12 and 9 days for the mid-April and mid-May SD
I think is more clear to give the information in terms of the days from sowing to emergence for the three SD
Response: We agree with your comment. Thank you.
Line 143
… growth clearly increased when sowing was delayed.
What do you mean by growth, plant height, forage yield? Please specify better
Response: In that case, we meant growth duration. There was a mistake in that sentence. We presented our results about yield/height in the next section 3.2. Thanks for your comment.
Lines 152-153
It would be better to give the plant height in meters (international unit), instead in cm. In Table 1 plant heigth is given in meters
Response: We agree. The text has been modified.
Line 158
Consequently, the amount of intercepted solar radiation during their respective periods of growth was also higher for the late SD
IPAR has been measured only once, I suppose that after silking, but no during the period of maize growth. Please, support this phrase, information is not in Table 4. It is form literature?
Response: The measurement was done about a week after silking. We have included that in the M&&M section. Thanks for your comment.
Line 169
March temperatures in the second and third growing seasons were below 10oC (Table 1), which affected the germination of the maize in the early SD, please add
Response: the sentence was not clear enough. We added “in the early SD”, thanks.
Lines 177-178
The final three-year average densities for the three SD were 72,000 plants ha-1, 79,000 plants ha-1 and 74,000 plants ha-1, for the early, middle and late SD, respectively.
Please specify in Mat&met how you measure this variable and if possible include the information for the 3 SD and the 2 hybrids in Table 4.
Response: Thanks for the comment. We have included plant density measurement in the M&M and in the table 4 as well.
Lines 179-180
The differences in LAI observed seem to confirm the need to increase the sowing density of early maturing hybrids
Data will recommend to increase the sowing density at early SD; there is no information of cultivars.
Please add in that phrase at early SD and remove early maturing hybrids
The differences in LAI observed seem to confirm the need to increase the sowing density at early SD
Response: You are right. We have changed the sentence. Thanks.
The hybrids used are early mature hybrids?
Response: No they are not. They are the traditional hybrids used as control for the cultivar studies.
Lines 189
differences in photoperiod of from 3 to 5 hours are needed during
Lines 124-149 and Lines 198 on
- Subheading 3.1
In Table 4. There is a significant interaction between V*SD and Y*V*SD in “days to emergence”, “days to silking” and “days to black layer”. You should comment on this if these interactions are qualitative you should consider the analysis for each hybrid separately.
Response: We have included this in the discussion. Thanks for pointing that out.
- Subheading 3.3
In addition, there is a significant V*SD interaction for grain yield and grain humidity, you should comment on this
Response: This has also been included in the discussion section. Thanks.
Reviewer 2 Report
See the attachment. My comments are given there. I was astonished by not finding presentation and discussion of cultivar results. They are extensively preseented in Table 4, the analysis of variance.

Author Response
Response to reviewer 2
See the attachment. My comments are given there. I was astonished by not finding presentation and discussion of cultivar results. They are extensively preseented in Table 4, the analysis of variance.
Response: Thanks for your comments, they have been valuable to improve our manuscript. The new manuscript is uploaded with track of changes.
We have included in the discussion the effect of the cultivar in the different measurements taken in the study. We did not included the cultivar effect because we thought that was important to report an overall effect rather than a particular cultivar. However, we agree with your comment and we have introduced some results and discussion about the cultivar when differences were found. Thanks.
Reviewer 3 Report
General comment:
The manuscript is merely descriptive and does not add to the knowledge in the area of agronomic science. Rather, it comes to conclusions which already have been recognised among agronomists since decades. Much of the text consists of notions which may appear trivial, for instance, that optimal sowing windows are not constant over the years and may vary according to the weather, or that highest yields generally occur where the growing season is longest and soil moisture is not a limiting factor. The ms. does not present any conclusion which has not been published before by others or by the authors. The contents of the ms. would better be suited to be published in a farmers' magazine rather than in a scientific journal. Since the experiments were done at a single location, generalisation of the observations remains quite limited. The key message seems to be that "the general perception of farmers that April sowings are the most appropriate in the area where the experiments were carried out" is corroborated by the findings presented in the manuscript.
Specific comments:
l. 58 "Although yield loss is still possible if...":
(1) Sounds kind of self-referential. If sowing really is "too" early (namely, too early for getting the maximum yield), then yield loss is not just a possibility; rather, it is a certainty.
(2) As noted by the authors in l. 30, optimum SD may vary from area to area. It remains obscure how planting-date windows recommended for Iowa (ref. 5) could suggest the existence of an earlier optimum planting window in Catalonia.
Table 4: Please check the averages for the SD's over the three yrs. (and, subsequently, the results from ANOVA). For instance, average grain yields seem to be 13.2, 14.0, and 12.8, rather than 12.8, 13.7, and 12.3, respectively. Check also average grain humidities.
Author Response
Response to reviewer 3
General comment:
The manuscript is merely descriptive and does not add to the knowledge in the area of agronomic science. Rather, it comes to conclusions which already have been recognised among agronomists since decades. Much of the text consists of notions which may appear trivial, for instance, that optimal sowing windows are not constant over the years and may vary according to the weather, or that highest yields generally occur where the growing season is longest and soil moisture is not a limiting factor. The ms. does not present any conclusion which has not been published before by others or by the authors. The contents of the ms. would better be suited to be published in a farmers' magazine rather than in a scientific journal. Since the experiments were done at a single location, generalisation of the observations remains quite limited. The key message seems to be that "the general perception of farmers that April sowings are the most appropriate in the area where the experiments were carried out" is corroborated by the findings presented in the manuscript.
Response: We disagree with the general comment. In fact, we consider the study as unique in the area of the study. As a general rule, maize is planted around mid-April based on farmer’s knowledge or studies from different areas. Our study will provide with results of the sowing effect on the maize development and yield. Thus, farmers and scientist could take decisions depending or their matter of interest.
Specific comments:
l. 58 "Although yield loss is still possible if...":
(1) Sounds kind of self-referential. If sowing really is "too" early (namely, too early for getting the maximum yield), then yield loss is not just a possibility; rather, it is a certainty.
Response: We agree with the comment and have rephrase the sentence. Thanks.
(2) As noted by the authors in l. 30, optimum SD may vary from area to area. It remains obscure how planting-date windows recommended for Iowa (ref. 5) could suggest the existence of an earlier optimum planting window in Catalonia.
Response: Thanks for the comment. We have decided to remove that sentence as it summarized the previous paragraph and therefore it was repetitive.
Table 4: Please check the averages for the SD's over the three yrs. (and, subsequently, the results from ANOVA). For instance, average grain yields seem to be 13.2, 14.0, and 12.8, rather than 12.8, 13.7, and 12.3, respectively. Check also average grain humidities.
Response: Thanks for pointing that out. We checked our original files and there was a mistake in the submitted table.
Round 2
Reviewer 1 Report
The authors have clarified the representativeness of the two cultivars, Cecilia - cycle 600 FAO and Eleonora cycle 700 FAO, used in the study. In addition, they have discussed the interaction between the two hybrids and the sowing dates in the duration of the growing cycle (section 3.1) and on the grain humidity (section 3.3).
The Materials and methods section has been reorganized and it is now easier to understand this section, the design of the experiment, its experimental conditions and the agronomic practices.
I had extensive comments on the first version of the manuscript; and all of these comments have been addressed satisfactorily in the revision. I consider the article is ready to be accepted for publication.
I am not a native English speaker, so I cannot tell about the correctness of English
Reviewer 3 Report
The specific comments have been considered in the revised version. However, the general impression of a quite limited (if any) scientific knowledge gain remains. A single trial location does not provide a sound basis for conclusions in agricultural research.